# Two Birds with One Stone: Protecting DNN Models Against Unauthorized Inference and Domain Transfer

## Abstract

Pre-trained Deep neural network (DNN) models are valuable intellectual property (IP) owing to their impressive performance, which might be extracted by attackers for unauthorized use. While existing protection schemes primarily focus on preventing attackers from obtaining high-performance models, another vulnerability regarding extracted models, i.e., the transferability has been largely under-explored, where attackers could transfer the model to another domain with good performance. Considering this dual vulnerability cannot be addressed by simply combining existing solutions for each problem, for the first time, this work jointly addresses these two security concerns and proposes DeTrans, a pre-trained model protection framework that utilizes bi-level optimization to protect weights of highly transferable filters, so as to prevent both unauthorized inference and cross-domain transfer followed by model extraction. Importantly, DeTrans ensures that the model functionality can be preserved for authorized users with specialized hardware support. The experiments demonstrate that DeTrans can significantly reduce accuracy on the source domain to random guessing and achieve up to an 81.23% reduction in transferability to the target domain.

## 1 Introduction

The success of on-device machine learning (ML) can be attributed to high-performance deep neural networks (DNNs) Dhar et al. (2021). Given the significant effort required for model training (i.e., massive data labeling and computational resources), pre-trained DNNs are valuable intellectual property (IP) assets for model owners Xue et al. (2021). However, model extraction has been demonstrated to be practical and successful for on-device ML, where the exact DNN models can be extracted via run-time attack Sun et al. (2021). Such attacks, constituting IP violation, potentially lead to significant losses for model owners. As a countermeasure, previous works primarily focus on preventing unauthorized inference for the victim models on their source domain Chakraborty et al. (2020); Chen et al. (2019); Zhou et al. (2023). However, these works have neglected another attack objective following model leakage, i.e., cross-domain transfer.

**Why are attackers motivated to conduct cross-domain transfer on the victim models?** In practice, model IP attackers are motivated to transfer the maliciously extracted model to other domains of interest, since they can exploit the common features learned by the pre-trained model to overcome data scarcity and save effort compared to training from scratch Zhuang et al. (2020). This is also an infringement of model owners' rights and has been identified in prior works Wang et al. (2022; 2023). Defense strategies have been suggested in these works, but they predominantly rely on training models from scratch to safeguard against transferability, which restricts their applicability to general pre-trained models. Moreover, employing these training strategies to obtain a pre-trained model may not be the optimal approach. This could potentially impact not only the model's accuracy but also other attributes (e.g., model owners lose transferability advantage themselves). Despite their inapplicability to pre-trained models, the models they produce consistently exhibit high performance on the source domain, making them vulnerable to unauthorized inference by attackers. Importantly, addressing this vulnerability is not as straightforward as merging them with methods employed to safeguard the source domain's performance, since these methods designed to tackle distinct issues utilize disparate training strategies and start from scratch Wang et al. (2022; 2023); Chakraborty et al.

(2020). Consequently, model owners cannot sequentially apply different methods to train the model and expect it to concurrently address both vulnerabilities.

Considering both unauthorized inference and cross-domain transfer are practical and common attack objectives (as illustrated in Fig. 1), we are motivated to propose a comprehensive protection method that can post-process pre-trained models to achieve (i) accuracy degradation on the source domain for unauthorized use (ii) reduced transferability to target domains. However, implementing such solutions are not as straightforward as combining known methods, but facing the following challenges. First, the potential target domains of attackers are uncertain. Second, we should implicitly measure the performance of possible model transfers during model protection, thereby ensuring the protected model remains robust against cross-domain transfers. Last, it is important to maintain high performance on the original domain for authorized users.

To address these challenges, we propose **DeTrans**, a novel approach that can provide double protection for pre-trained DNN models. For the first challenge, while we cannot predict the specific target domain, we make a reasonable assumption that the distribution of the potential target domain is likely to be close to that of the source domain, motivated by the threat model of cross-domain transfer attack Torrey & Shavlik (2010). With this assumption, we generate auxiliary domains based on the source domain as inspired by single domain generalization to solve the target domain uncertainty Qiao et al. (2020); Volpi et al. (2018). To address the second challenge, we formulate a bi-level optimization problem, with the lower level assessing the model transferability.

Moreover, DeTrans leverages hardware support in the form of a trusted execution environment (TEE) Ngabonziza et al. (2016); Truong et al. (2021) to address the last challenge. Specifically, DeTrans only optimizes the weights of the filter with the highest transferability in each convolutional layer of the victim DNN model. Since these weights capture the common features between the source and potential target domains, perturbing weights associated with these features can degrade performance on both domains. Therefore, when deploying the protected model on device, attackers cannot achieve high performance on both domains, thus achieving double protection. Meanwhile, the original value of the perturbed weights is small enough to be stored in the stringent secure memory of TEE Ngabonziza et al. (2016), which can be used to restore the high performance on the source domain for users. The contributions of this work are summarized as follows:

- To the best of our knowledge, this is the first work that mitigates the risk of model leakage by preventing attackers from both unauthorized inference and cross-domain transfer, thus achieving comprehensive protection for pre-trained DNN models.

- Leveraging the generated auxiliary domains, DeTrans optimizes critical weights of the victim model through a bi-level optimization. The resulting protected model exhibits nearly random guessing performance for attackers on the source domain, and reduces transferability by up to 81.23% on potential target domains.

- Experiments demonstrate DeTrans outperforms the state-of-the-art model protection works and exhibits robustness against different fine-tuning methods employed by attackers.

## 2 BACKGROUND

### 2.1 THREAT MODEL

**Attackers.** To achieve comprehensive protection, we assume a strong model IP attacker who has knowledge of the domain in which the victim model is trained. We also assume the attacker is able to extract the DNN model stored in unsecured memory including its architecture and pre-trained weights, e.g., using the run-time attack Sun et al. (2021). As illustrated in Fig. 1, the attackers could exploit the extracted model in two ways: **1)** they directly use the model for **unauthorized inference** when their interests aligns with source domain; **2)** they fine-tune the extracted model to achieve **cross-domain transfer**

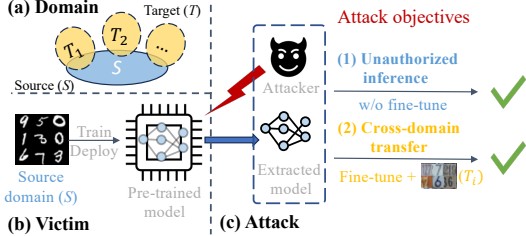

Figure 1: Attack model.

when they collect *a small set of data* from the target domain, with the model architecture remaining unchanged.

**Model IP owners.** To protect pre-trained models against illegal model exploitation while preserving the model performance for authorized users, model owners could employ defense solutions with hardware support (e.g., TEE), which can block the access to critical data stored in secure memory. Specifically, owners will generate a protected model to be deployed on devices, and crucial data (the original values of the perturbed weights) is safeguarded by TEE. During inference, users can access this data but cannot extract it, thanks to the properties of TEE Chakrabarti et al. (2020). Besides, it is worth noting that our intention is not to block *all* potential target domains, as this would be impractical given the vast array of possibilities. Instead, we consider a common and practical scenario that the target domains are close to the source domain, while sharing the same label space.

## 2.2  SINGLE DOMAIN GENERALIZATION

To protect models from illegal cross-domain transfer, we face the challenges that the target domains are unknown during model protection. This challenge also exists in single domain generalization Zhou et al. (2022), where its objective is to allow the model learned from a single source to be able to generalize over a series of unknown distributions. Furthermore, it also assumes that the unknown target domain is close to the source, which aligns with our assumption. A common practice in single domain generalization is to augment the source domains to approximate the unknown domain distributions Qiao et al. (2020); Volpi et al. (2018); Fan et al. (2021). For example, Volpi *et al.* propose an adaptive data augmentation method, which adds adversarially perturbed samples to the data space Volpi et al. (2018). Also, Qiao *et al.* generate fictitious domains using a Wasserstein Auto-Encoder to encourage large domain discrepancy in the input space Qiao et al. (2020).

Given the shared challenge and assumption, we generate auxiliary domains by augmenting the source domain using similar methods. However, it should be noted that our design objective is opposite of domain generalization, i.e., we intentionally reduce the victim model's generalization ability, thus impeding its ease of transfer to unknown target domains.

## 2.3  TRUSTED EXECUTION ENVIRONMENT

TEE (e.g., ARM TrustZone Ngabonziza et al. (2016)) has been used to protect model protection by creating hardware-based isolation between the secure and normal memory of a device to isolate sensitive information Sun et al. (2023); Chakraborty et al. (2020); Chen et al. (2019); Zhou et al. (2023). However, the secure memory space is relatively limited, e.g., 3-5MB in TrustZone Amacher & Schiavoni (2019), compared to the size of normal DNNs Chen et al. (2019). To address this issue, we aim to identify a subset of critical model weights that affect the model performance for both the source domain and the target domains. Without authorized access to the critical weights stored in TEE, the extracted model part will perform poorly due to a lack of critical data. Additionally, it is noted that even with authorized inference, the data inside the TEE remains a black box for authorized users. While they can leverage the data within the TEE to achieve high inference accuracy on the source domain, they cannot extract the data (i.e., critical model weights in our case). Since the effectiveness of TEE has been previously established in other studies Chen et al. (2019); Sun et al. (2023), this work will not delve into the implementation details.

## 2.4  DISCUSSION ON RELATED WORKS

As model IP protection is critical for model owners, several studies have explored methods to degrade the model's performance to mitigate the risk of model leakage. For example, Chakraborty *et al.* propose a key-dependent training algorithm to obfuscate the weight space such that the accuracy of the model drops significantly when it is extracted and used on different devices Chakraborty et al. (2020). Moreover, Zhou *et al.* introduce a protection approach for pre-trained models, which obfuscates a small set of model weights using a fine-grained mask designed by a reinforcement learning-based controller Zhou et al. (2023). However, these methods primarily focus on degrading accuracy on the source domain, which cannot naturally reduce the transferability to prevent cross-domain transfer by attackers.

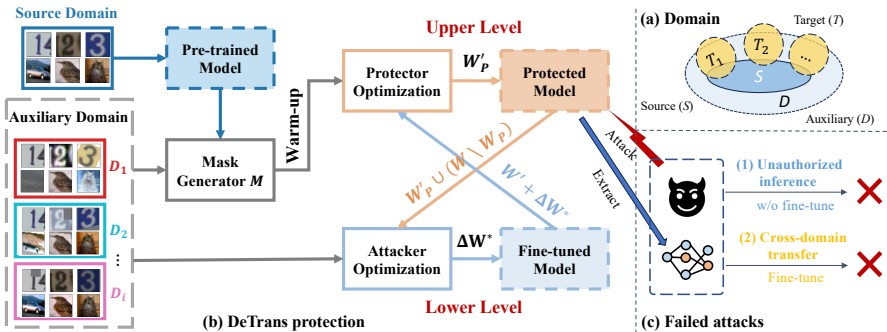

Figure 2: The framework overview of DeTrans. DeTrans leverages bi-level optimization to protect the pre-trained model for both accuracy and transferability reduction. The resulting protected model can prevent attackers from unauthorized inference and cross-domain transfer.

Taking transferability into account, Wang *et al.* propose a training strategy to restrict model transferability by capturing exclusive data representation in the source domain Wang et al. (2022). Furthermore, the compact un-transferable isolation domain is proposed to prevent cross-domain transfers by emphasizing the private style features of the authorized domain Wang et al. (2023). However, these methods cannot be extended to general pre-trained models, and using such methods to train a model will naturally restrict the model's transferability, even for model owners themselves. More importantly, Wang et al. (2022; 2023) are designed for machine-learning-as-a-service (MLaaS), where the service providers have to guarantee good performance for users on the source domain. As a result, the preserved high performance renders them ineffective in preventing attackers from making unauthorized inferences.

## 3    PROPOSED FRAMEWORK: DETRANS

Aiming at achieving the best for preventing both illegal usages, i.e., unauthorized inference and cross-domain transfer, we propose an extensive model IP protection scheme that can provide accuracy degradation on the source domain and transferability reduction on the target domain at the same time. The workflow of the proposed method, referred to as DeTrans, is illustrated in Fig. 2. We first generate the auxiliary domains by augmenting the source domain, then select critical weights via a filter-wise mask to fit the small secure memory. Later, we obtain the protected model by modifying the selected weights via a bi-level optimization process. The modified weights hinder attackers from leveraging the victim model and successfully protect model IP.

### 3.1    AUXILIARY DOMAINS

We introduce a generative domain augmentation method to generate auxiliary domains $\mathcal{D}_i$ based on the source domain. The rationale behind this approach is that attackers are more likely to opt for model transfer, when the target domain closely aligns with the source domain. This choice provides them with the advantages of transfer learning while avoiding potential drawbacks, as discussed in Pan & Yang (2010); Wang et al. (2019b). Therefore, even though we cannot precisely predict the specific target domain that attackers may select, we can approximate its distribution by augmenting the source domain.

Inspired by Qiao et al. (2020), we build a Wasserstein Auto-Encoder (WAE) parameterized by $\theta$ to introduce domain discrepancy between the source domain $\mathcal{S}$ and the auxiliary domains $\mathcal{D}_i$. However, to simulate cross-domain transfer, we need to generate labeled samples, simulating how attackers use limited labeled data to fine-tune the extracted model. To this end, we enhance the WAE by incorporating the label as a condition in the decoding phase. Specifically, it consists of an encoder $Q(x|z)$ and a decoder $G(z, y)$, where $x$ and $y$ represent the input and its label, and $z$ represents the bottleneck embedding. Additionally, we employ Maximum Mean Discrepancy (MMD) to quantify the distance between $Q(x)$ and the Gaussian prior $P(z)$, following the approach in Tolstikhin et al. (2018). Detailed implementation specifics can be found in Appendix A. The conditional WAE is

optimized according to:

$$\min_{\theta} ||G(Q(x), y)) - x||_2 + \text{MMD}(Q(x), P(z)) \tag{1}$$

After optimization, we can generate samples to construct auxiliary domains via the decoder $G$. To encourage a range of domain discrepancies that better align with the unseen target distribution, we introduce feature augmentation by incorporating a latent vector $h$ sampled from $\mathcal{N}(\mu, \sigma)$ into the feature embedding, as outlined in Qiao & Peng (2021). This augmentation technique has been proven effective in enhancing domain discrepancy through variations in $\mu$ and $\sigma$. As a result, we can generate diverse auxiliary domains $\mathcal{D}_i$. Each auxiliary domain is then divided into a training set and a validation set, serving as components for the lower-level optimization (Eq. 5) and upper-level optimization (Eq. 6), respectively.

## 3.2 MASK DESIGN

To overcome the limitations of secure memory in the Trusted Execution Environment (TEE), DeTrans introduces a filter-wise mask design, allowing for the modification of weights exclusively for selected filters. The filter selection strategy centers on identifying the filter with the highest transferability in each layer. This selection is based on the understanding that filters with higher transferability capture shared patterns between domains, making them more intrinsically transferable than others. By adjusting the weights of these high-transferability filters, we disrupt the shared features learned by both the source and target domains, thus achieving concurrent accuracy degradation and transferability reduction. Since the target domains are unknown in our case, we replace them with auxiliary domains.

In quantifying filter transferability, we adopt the approach used in Wang et al. (2019a). We first measure the discrepancy $d$ between the source domain $\mathcal{S}$ and each auxiliary domain $\mathcal{D}_i$ for each channel $c$ (output of a filter) following Eq. 2, where we omit $i$ for simplicity.

$$d^{(c)} = \left| \frac{\mu_{\mathcal{S}}^{(c)}}{\sqrt{\sigma_{\mathcal{S}}^{2\,(c)}}} - \frac{\mu_{\mathcal{D}}^{(c)}}{\sqrt{\sigma_{\mathcal{D}}^{2\,(c)}}} \right|, \text{ where } \mu^{(c)} = \frac{1}{b} \sum_{j=1}^{b} \boldsymbol{x}_j^{(c)}, \; \sigma^{2\,(c)} = \frac{1}{b} \sum_{j=1}^{b} (\boldsymbol{x}_j^{(c)} - \mu^{(c)}). \tag{2}$$

Here, $b$ denotes the batch size and $\boldsymbol{x}$ represents the feature map of convolutional layers. We choose $\boldsymbol{x}$ from source domain when calculating $\mu_{\mathcal{S}}$ and $\sigma_{\mathcal{S}}^2$, and from $\mathcal{D}_i$ otherwise. Hence, the distance-based probability $\boldsymbol{\alpha}$ for each channel c is calculated as:

$$\boldsymbol{\alpha}^{(c)} = \frac{K(1 + d^{(c)})^{-1}}{\sum_{n=1}^{K} (1 + d^{(n)})^{-1}}, \quad c = 1, 2, ..., K \tag{3}$$

where $K$ is the channel size, and a higher $\boldsymbol{\alpha}$ score indicates greater transferability of a filter. We calculate the $\boldsymbol{\alpha}$ scores for each source-auxiliary domain pair and take the average $\boldsymbol{\alpha}$ to rank the filters. Consequently, a binary mask $\mathbf{M}$ is generated, with 1 indicting the selected filter with the highest transferability in each layer.

## 3.3 BI-LEVEL OPTIMIZATION

We consider a pre-trained model $\mathcal{M}$ with weights $\mathbf{W}$ as the victim model, which is trained on labeled data pairs $(\boldsymbol{x_s}, \boldsymbol{y_s})$ from the source domain $\mathcal{S}$. We selectively modify a small set of weights $\mathbf{W_P}$ corresponding to the chosen filters, while keeping the rest fixed as $\mathbf{W} \setminus \mathbf{W_P}$. A basic protection method is adding weight changes $\Delta \mathbf{W_P}$ to $\mathbf{W_P}$ to degrade the performance (i.e., accuracy drop), thus preventing attackers from unauthorized inference. The optimization of modified weights $\mathbf{W_P'}$ (:= $\mathbf{W_P} + \Delta \mathbf{W_P}$) follows:

$$\max_{\mathbf{W_P'}} \mathcal{L}\left(f\left(\boldsymbol{x_s}; \mathbf{W_P'} \cup (\mathbf{W} \setminus \mathbf{W_P})\right), \boldsymbol{y_s}\right), \tag{4}$$

where $f$ denotes the functionality of the DNN, and $\mathcal{L}$ denotes the loss function of the source task.

Given that our protection extends beyond unauthorized inference to include preventing cross-domain transfer, we improve the above method by formulating a bi-level optimization to achieve our goal. This framework enables us to comprehensively address the model protection challenge, alternating stages while implicitly considering the transferability of the protected model through the use of auxiliary domains.

**Lower-level Optimization**. In this stage, we emulate how attackers optimize the extracted model for transfer by finding an optimal weight update $\Delta \mathbf{W}$ on the extracted model to enhance performance in the target domain. This mimicking is done using training data $(\boldsymbol{x}_i^{tr}, \boldsymbol{y}_i^{tr})$ in the auxiliary domain $\mathcal{D}_i$. We optimize $\Delta \mathbf{W}$ with the following equation:

$$\Delta \mathbf{W}_i^* = \arg \min_{\Delta \mathbf{W}_i} \mathcal{L} \left( f \left( \boldsymbol{x}_i^{tr}; \mathbf{W}' + \Delta \mathbf{W}_i \right), \boldsymbol{y}_i^{tr} \right) \tag{5}$$

where $\mathbf{W}'$ is the weights of the protected model obtained from the upper-level optimization.

**Upper-level Optimization**. To leverage the knowledge gained from the lower-level optimization regarding the transferability of the model with current weights $\mathbf{W}'$, we proceed to optimize the weights $\mathbf{W}'$ further. The goal here is to diminish the transferability of the model across all auxiliary domains. This process is guided by the following equation:

$$\max_{\mathbf{W}_{\mathbf{P}}'} \sum_{i=1}^{\mathcal{I}} \lambda_i \mathcal{L} \left( f \left( \boldsymbol{x}_i^{val}; \mathbf{W}' + \Delta \mathbf{W}_i^* \right), \boldsymbol{y}_i^{val} \right), \; s.t. \; \mathbf{W}' = \mathbf{W}_{\mathbf{P}}' \cup (\mathbf{W} \setminus \mathbf{W}_{\mathbf{P}}) = \mathbf{W}_{\mathbf{P}}' \odot \mathbf{M} + \mathbf{W} \odot (\mathbf{1} - \mathbf{M}) \tag{6}$$

where $\lambda_i$ is the scaling factor of the loss on $(\boldsymbol{x}_i^{val}, \boldsymbol{y}_i^{val})$, i.e., the samples and labels from the validation set in the auxiliary domain $\mathcal{D}_i$ and $\sum_{i=1}^{\mathcal{I}} \lambda_i = 1$. Notably, we also incorporate the source domain into the auxiliary domain during the bi-level optimization to ensure accuracy degradation on the source domain.

## 3.4 SUMMARY OF DETRANS

Integrating all the aforementioned strategies, we present DeTrans in Algorithm 1, with the following key points: (i) Given a pre-trained model, DeTrans identifies the most critical filters for transfer learning and generates a corresponding mask $\mathbf{M}$ for optimization; (ii) DeTrans implements basic protection as a warm-up to initialize $\mathbf{W}_{\mathbf{P}}'$; (iii) DeTrans employs a bi-level optimization to further optimize $\mathbf{W}_{\mathbf{P}}'$. Here we also add $\mathcal{S}$ as one $\mathcal{D}_i$ during bi-level optimization to ensure that accuracy degradation is maintained on the source domain. The optimal protected model will be generated upon convergence, which could prevent attackers from malicious exploitation, including direct use and model transfer. Besides, the protected original weights $\mathbf{W}_{\mathbf{P}}$ are stored in TEE secure memory, which corrects the functionality of the protected model for authorized users.

## 4 EXPERIMENTS

We evaluate the performance of DeTrans on multiple DNN models and datasets for the classification tasks, demonstrating its effectiveness compared to other methods. As DeTrans targets protecting on-device DNN models, where the memory and computational capability is limited compared to cloud ML, we examine DeTrans on relatively small DNNs and datasets in line with the actual situation. Additionally, we conduct ablation studies to examine the influence of important modules in DeTrans.

## 4.1 EXPERIMENT SETUP

**Dataset.** We examine our methods on two types of datasets following previous works on non-transferable learning Wang et al. (2022; 2023). One is the **digits datasets**, including *MNIST (MN)* Lecun et al. (1998), *USPS (US)* Hull (1994), and *SVHN (SV)* Netzer et al. (2011). The training and test sets for these digit datasets are unbalanced, i.e., the number of each class is unequal. The other is the **natural image datasets**, including CIFAR10 *(CF10)* Krizhevsky et al. (2009) and STL10 Coates et al. (2011). The details of the dataset and data processing are shown in Appendix B.

**DNN Models.** We train VGG-11 Simonyan & Zisserman (2015) and ResNet-18 He et al.

---

**Algorithm 1** DeTrans

**Input**: pre-trained model $\mathcal{M}$; auxiliary domains $\mathcal{D}_i$
**Output**: protected model $\mathbf{W}'$

    // Generate mask
1: Calculate $\boldsymbol{\alpha}^{(c)}$ across $\mathcal{D}_i$     ▷ Eq. (3)
2: Generate a filter-wise mask $\mathbf{M}$
    // Apply basic protection as a warm-up
3: Initialize $\mathbf{W}_{\mathbf{P}}'$ with $\mathbf{M}$ constrained     ▷ Eq. (4)
    // Bi-level optimization loop
4: **repeat**
5:     $\mathbf{W}' \leftarrow \mathbf{W}_{\mathbf{P}}' \cup (\mathbf{W} \setminus \mathbf{W}_{\mathbf{P}})$
6:     **for** all $\mathcal{D}_i$ **do**
7:         Obtain $\Delta \mathbf{W}_i^*$ in lower-level stage  ▷ Eq. (5)
8:         Calculate loss with $\mathbf{W}' + \Delta \mathbf{W}_i^*$
9:     **end for**
10:    Update $\mathbf{W}_{\mathbf{P}}'$ in upper-level stage    ▷ Eq. (6)
11: **until** Converge

---

Table 1: The results of defending against unauthorized inference, reported with the top-1 accuracy (%) on test dataset. Here the baseline accuracy is the benign accuracy of victim models.

| Domain | VGG-11 | | | | | | ResNet-18 | | | |
|---|---|---|---|---|---|---|---|---|---|---|
| | Baseline | *SL* | *NTL* | *CUTI* | *NNSplitter* | **DeTrans** | Baseline | *SL* | *NNSplitter* | **DeTrans** |
| MN | 99.03 | 79.18 | 97.9 | 99.1 | 9.90 | 10.04±0.29 | 99.50 | 92.14 | 9.99 | 10.01±0.23 |
| US | 95.61 | 17.89 | 98.8 | 99.6 | 12.31 | 13.25±1.22 | 96.51 | 57.55 | 8.78 | 9.04±0.41 |
| SV | 93.57 | 20.12 | 88.4 | 90.9 | 8.87 | 8.90±2.45 | 94.28 | 51.42 | 8.15 | 8.81±1.52 |
| CF10 | 92.39 | 42.61 | / | / | 10.00 | 10.00±0.00 | 93.07 | 42.93 | 10.00 | 10.00±0.00 |
| STL10 | 78.44 | 11.25 | / | / | 10.00 | 10.00±0.00 | 79.12 | 11.67 | 10.00 | 10.00±0.00 |

(2016) on different datasets as the victim models.
We select these two DNNs to match the victim
models in previous works Wang et al. (2022); Zhou et al. (2023). For fairness, we resize all images from different datasets into the same size to avoid the architecture changing (see Appendix C) when applying transfer learning. The victim models are trained using the Adam optimizer Kingma & Ba (2015).

**Settings.** We generate auxiliary domains based on the source domain, where $\mu$ is from 0 to 1 with step of 0.25 and $\sigma$ is set to 0.5 and 1. The generated examples are shown in Appendix D. Note that the target domain remains unseen during model protection. We assume attackers hold 5% of training data in the target domain to conduct cross-domain transfer. Also, the ratio of modified parameters is unchanged, i.e., 0.07% and 0.18% of total parameters for VGG-11 and ResNet-18, respectively. The corresponding storage size is 79KB and 122KB, which can meet the secure memory constraint on TEE. The hyperparameter settings can be found in Appendix E. We use top-1 inference accuracy (%) as the evaluation metric for all experiments, which are implemented using PyTorch and conducted on a machine equipped with an NVIDIA A10 GPU.

## 4.2 COMPARISON METHODS

As DeTrans is the first approach to simultaneously prevent unauthorized inference and cross-domain transfer (i.e., dual protection), there are no previous works fully comparable to it. However, to better evaluate the effectiveness of DeTrans, we adopt the following state-of-the-art (SOTA) works as a comparison on a single aspect. For preventing unauthorized inference, we select the SOTA method *NNSplitter Zhou et al. (2023)*, which shares the same objective as the basic protection mentioned in our work, and measure its performance on datasets in our work. As for defending against model transfer, the recent works *NTL Wang et al. (2022)* and *CUTI Wang et al. (2023)* can be used for comparison. Since they did not experiment with ResNet-18, we only include comparison on VGG-11. Moreover, we also propose *Baseline*, where no protection is applied to the victim model, and *Supervised Learning (SL)*, i.e., training from scratch with limited data (5% of training data).

## 4.3 PERFORMANCE EVALUATION

Based on our threat model, we consider two attack scenarios and demonstrate that DeTrans can defend against two attacks at the same time, while SOTA works can only achieve one aspect.

**Scenario I: Unauthorized Inference.** When attackers' target domains align with the source domain of the victim model, they can directly utilize the extracted model to conduct inference without further fine-tuning. To prevent attackers from such attack, DeTrans aims to degrade the accuracy on the source domain. By intentionally introducing modifications to the model weights, DeTrans ensures that the extracted model performs poorly and becomes ineffective for direct use by attackers.

From the results in Tab. 1, we can observe that by applying DeTrans, the top-1 inference accuracy experiences a significant drop compared to the baseline (i.e., from over 90% to ∼10%), where the baseline applies no protection to the victim model, allowing attackers to achieve the exact inference accuracy as the victim model. Besides, the succeed of DeTrans in accuracy drop results from misclassifying all images into a random class. Specially, since all classes in CF10 and STL10 are balanced, the accuracy is degraded to 10% with no variance.

Table 2: The results of defending against cross-domain transfer. The left of ⇒ is the source domain, and the right side is the target domain (unseen during protection). The average drop (Avg. Drop) is calculated by the average difference from the baseline, and the higher drop indicates better protection.

| | | MN ⇒ US | MN ⇒ SV | US ⇒ MN | US ⇒ SV | SV ⇒ MN | SV ⇒ US | Avg. Drop |
|---|---|---|---|---|---|---|---|---|
| | Baseline | 93.35 | 81.53 | 93.20 | 81.97 | 94.78 | 94.67 | / |
| | *SL* | 17.89 | 20.12 | 79.18 | 20.12 | 79.18 | 17.89 | 50.86 |
| VGG-11 | *NTL* | 13.8 | 20.8 | 6.7 | 6.0 | 12.3 | 8.9 | **78.50** |
| | *CUTI* | 6.7 | 6.7 | 9.1 | 25.5 | 11.9 | 14.3 | 77.55 |
| | *NNSplitter* | 92.67 | 80.52 | 93.11 | 81.71 | 94.34 | 94.40 | 0.46 |
| | **DeTrans** | 17.88±0.23 | 19.58±0.47 | 11.35±0.49 | 19.59±0.38 | 10.10±0.76 | 13.15±0.39 | 74.65 |
| | Baseline | 95.24 | 84.64 | 96.24 | 83.75 | 97.46 | 96.22 | / |
| ResNet-18 | *SL* | 57.55 | 51.42 | 92.14 | 51.42 | 92.14 | 57.55 | 25.22 |
| | *NNSplitter* | 94.98 | 83.82 | 95.87 | 82.85 | 97.54 | 96.17 | 0.39 |
| | **DeTrans** | 17.89±0.39 | 6.70±1.10 | 9.80±0.65 | 6.69±0.68 | 9.58±0.23 | 15.52±0.24 | **81.23** |

Moreover, it is noted that DeTrans can achieve a comparable degraded accuracy as the SOTA work *NNSplitter*. The degraded accuracy is even lower than the victim DNN trained with supervised learning (*SL*), rendering the unauthorized inference useless. In contrast, *NTL* Wang et al. (2022) and *CUTI* Wang et al. (2023) designed for restraining model transferability fail in preventing unauthorized inference, since their accuracy on the source domain are still high.

**Scenario II: Model Transfer.** We form several domain pairs to demonstrate that the models under DeTrans protection have reduced transferability. We provide results in Tab. 2 and Appendix F. As shown in Tab. 2, although *NNSplitter* shows the optimal performance in defend against unauthorized inference, it demonstrates almost no reduction in transferability compared to the baseline, with only 0.46% and 0.39% accuracy drop on VGG-11 and ResNet-18, respectively. The results underscores that accuracy reduction on the source domain does not necessarily impact the model transferability.

In contrast, DeTrans proves to be effective in reducing transferability, e.g., up to 81.23% accuracy drop on ResNet-18. Besides, it can achieve comparable protection as SOTA works (i.e., *NTL* and *CUTI*) on transferability reduction, with the difference less than 4%. More importantly, unlike *NTL* and *CUTI*, DeTrans does not require additional input-level modification to achieve transferability reduction, i.e., adding a spacial patch on the input sample.

Moreover, when attackers attempt to transfer the model to a more complex domain, such as MN ⇒ SV, the transferred accuracy is relatively low even without any protection (i.e., 81.53% for baseline). The complexity of SV is raised from its enriched feature and unbalanced class compared to MN Lorena et al. (2019). However, attackers may still be motivated to extract the model and transfer it to the target domain, as training a model from scratch with limited data would yield lower accuracy (i.e., 20.12% for *SL*). Therefore, reducing transferability in this case is still crucial. As shown in Tab. 2, the model under DeTrans protection achieve performance even lower than *SL*, making attackers' effort in model transfer useless.

Overall, DeTrans achieves comprehensive model protection by defending against unauthorized inference and model transfer at the same time, filling the gap of SOTA methods. It successfully reduces the accuracy on the source domain to the random-guessing level (∼10% for ten-class classification tasks), while restraining the performance on unseen target domains lower than attackers training one from scratch using *SL*.

### 4.4 Ablation Studies

**Filter Selection.** The filter selection strategy in the mask design of DeTrans is based on the transferability measurement of each channel (Sec. 3.2). To measure the influence of this strategy, we replace it with a random filter selection strategy, i.e., randomly selecting a channel to design a mask for model protection. We conduct the comparison experiments on the VGG-11 model, and the results are shown in Fig. 3, where the performance of

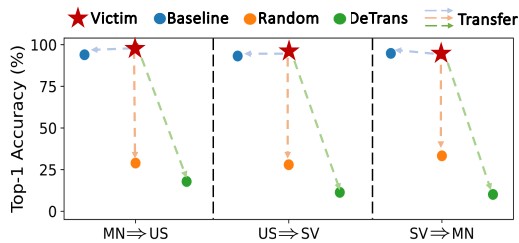

Figure 3: The comparison of random filter selection filter selection with DeTrans. The dot marks showing the accuracy of victim model after being transferred to the target domain.

victim is measured on the source domain. Overall, random filter selection can also reduce transferability compared to the baseline among all cases, but DeTrans performs even better with an extra 14.95% accuracy drop on average.

**Optimization Method.** Our bi-level optimization takes the potential attack surface (i.e., malicious model transfer) into consideration during model protection. To demonstrate its effectiveness, we assume a stronger protector who: (i) has knowledge of the target domain as the attacker (i.e., a small set of data in the target domain); (ii) and applies a *naive* model protection scheme, in which we assume the protector and attacker conduct "*defense-attack-defense-...*", in an alternative optimization manner as described in Eq. (7) and Eq. (8):

$$(\text{attack}) \ \Delta \mathbf{W}_a^* = \arg \min_{\Delta \mathbf{W}_a} \mathcal{L} \left( f \left( \boldsymbol{x_t}; \mathbf{W}' + \Delta \mathbf{W}_a \right), \boldsymbol{y_t} \right) \tag{7}$$

$$(\text{defense}) \ \max_{\mathbf{W}_{\mathbf{P}}'} \mathcal{L} \left( f \left( \boldsymbol{x_t}; \mathbf{W}_{\mathbf{P}}' \cup (\mathbf{W} \setminus \mathbf{W_P}) + \Delta \mathbf{W}_a^* \right), \boldsymbol{y_t} \right) \tag{8}$$

where $\Delta \mathbf{W}_a^*$ is the weights modification by attackers via transfer learning, $\mathbf{W}_{\mathbf{P}}'$ is optimization from the protector, and $(\boldsymbol{x_t}, \boldsymbol{y_t})$ denoted 5% labeled data from the target domain.

The model performance after 50-step naive optimization is reported in Tab. 3. Compared to the baseline method, the naive optimization only shows 1.3 % accuracy drop, while the bi-level optimization achieves over 80% accuracy drop. The reason behind this difference lies in the nature of the naive optimization process. As described in Appendix G, the naive optimization is dynamic, meaning that due to the same set of samples $(\boldsymbol{x_t}, \boldsymbol{y_t})$ but with opposite objectives in the two equations, each equation can counteract the modifications made by the other in the last step. In other words, the influence of one equation can be nullified by the other equation, resulting in a minimal impact on

Table 3: Comparison of transferability reduction on ResNet-18 model.

|  | MN⇒SV | US⇒MN | SV⇒US | **Avg. Drop** |
|---|---|---|---|---|
| Naive | 83.76±1.06 | 95.67±1.17 | 93.77±0.41 | 1.3 |
| Bi-level | 6.70±1.10 | 9.80±0.65 | 15.52±0.24 | **81.69** |

the model's accuracy. As the attacker is the one who makes the final modification on the extracted model, they can always improve the accuracy to a level comparable to the baseline. This explains the limited effectiveness of the naive optimization approach in reducing model accuracy compared to the bi-level optimization approach.

## 5 DISCUSSION

### 5.1 MODIFICATION AND FINE-TUNING RANGE

We explore different modification ranges for protectors and model fine-tuning ranges for attackers. For protectors, the choices include modifying: p1) the first layer, and p2) one filter per layer. For attackers, they can either conduct unauthorized inference on the source domain without fine-tune or fine-tune a1) the last layer or a2) the whole extracted model for transferring to the target domain. We evaluate the performance of each modification and fine-tuning range on the VGG-11 model trained on MN. The results

Table 4: The comparison of different modification and fine-tuning ranges, where the source domain is MN and the target is US.

|  | Protector | Attacker | DeTrans |
|---|---|---|---|
| w/o fine-tune (Source) | p1 | - | 12.59±3.60 |
|  | p2 | - | 10.04±0.29 |
| w/ fine-tune (Target) | p1 | a1 | 58.07±0.43 |
|  | p1 | a2 | 39.36±0.44 |
|  | p2 | a1 | 22.27±0.33 |
|  | p2 | a2 | 17.88±0.23 |

in Tab. 4 show that p2 outperforms p1 when defending against the unauthorized inference. Besides, Tab. 4 also presents the performance of each combination when the attacker aims to transfer the model to another domain (i.e., US). When adopting p2, attackers can only achieve low accuracy ($\sim$20%) no matter they fine-tune the last layer or the whole model. The results demonstrate the advantage of spreading the weights modification across all layers.

### 5.2 APPLICABILITY BEYOND SMALLER MODELS

Since this works focuses on securing the IP of on-device DNN models, which has been demonstrated as vulnerable to model extraction attacks Sun et al. (2021), thus, we choose models with fewer layers as the victim models for above experiments and demonstrate the effectiveness of

Table 5: The performance of cross-domain transfer on CF10.

| Target | Baseline | NTL | CUTI | **DeTrans** |
|--------|----------|-----|------|-------------|
| STL10 | 58.0 | 14.8 | 14.0 | **10.5** |

DeTrans. However, as a generic defense strategy, our proposed methodology can also be applied to larger models with more layers, if the device can accommodate the model. We use ResNet-50 as a case study, which was used as largest model in previous related works Wang et al. (2022; 2023). We obtain a pre-trained ResNet-50 on CIFAR-10 with inference accuracy of 93.65%. By applying DeTrans protection with 0.14% of weights being modified, the accuracy of unauthorized inference degrades to 10%. Meanwhile, as shown in Tab. 5, the cross-domain transfer only achieves 10.5% with DeTrans, which also outperforms SOTA in reducing transferability.

## 6 CONCLUSION

In this work, we introduce DeTrans, a novel framework for protecting on-device DNN models against unauthorized inference and cross-domain transfer while preserving model performance for users using TEE. By selectively modifying a small subset of weights in the pre-trained model, DeTrans achieves near-random guess performance on the source domain and transferability reduction for potential target domains (> 80% for ResNet-18). The original values of these modified weights are stored in TEE, which is only accessible to authorized users. Moreover, DeTrans remains effective regardless of how attackers fine-tune the protected model, and it is applicable to larger models. Overall, it fills the gap of SOTA model protection works, which either fail to prevent unauthorized inference or overlook the cross-domain transfer from attackers.

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

## A  IMPLEMENTATION OF WAE

The design of the encoder $Q$ and the decoder $G$ follows the prior work Tolstikhin et al. (2018). The input size of $Q$ is $32 \times 32 \times 1$ for digit datasets and $32 \times 32 \times 3$ for CIFAR10 & STL10. Specifically, the encoder architecture is (using input size of $32 \times 32 \times 3$ as an example):

$$x \in \mathbb{R}^{32 \times 32 \times 3} \to \mathrm{Conv}_{128} \to \mathrm{BN} \to \mathrm{ReLU}$$
$$\to \mathrm{Conv}_{256} \to \mathrm{BN} \to \mathrm{ReLU}$$
$$\to \mathrm{Conv}_{512} \to \mathrm{BN} \to \mathrm{ReLU}$$
$$\to \mathrm{Conv}_{1024} \to \mathrm{BN} \to \mathrm{ReLU} \to \mathrm{FC}_{64},$$

and the decoder architecture is:

$$z \in \mathbb{R}^{74} \to \mathrm{FC}_{4 \times 4 \times 1024}$$
$$\to \mathrm{ConvTranspose}_{512} \to \mathrm{BN} \to \mathrm{ReLU}$$
$$\to \mathrm{ConvTranspose}_{256} \to \mathrm{BN} \to \mathrm{ReLU}$$
$$\to \mathrm{ConvTranspose}_{128} \to \mathrm{BN} \to \mathrm{ReLU} \to \mathrm{FSConv}_3,$$

where ConvTranspose denotes transposed convolutional layers. Here the input of decoder is the concatenation of the output latent vector of $Q$ (64-dimension) and the one-hot encoding of the 10-class labels (10-dimension).

Both architectures include 4 convolutional layers with the filter size of $5 \times 5$. The stride is set to 2 for the first 3 Conv and ConvTranspose in $Q$ and $D$, respectively. The WAE is optimized for 100 epochs according to Eq. 1 using the Adam optimizer. The learning rate is initially set to 0.001 and decreased by factor of 2 after 20 epochs.

## B  DETAILS OF DATASETS

**Digits.** *MNIST* contains 70k gray-scale images of handwritten digits from 0 to 9, each of which is $28 \times 28$ pixels in size. The training set includes 60k labeled data and the rest data are for testing. *USPS* has 7291 training and 2007 test images in gray-scale with size $16 \times 16$. It was created by taking a sample of real-world handwritten digits from envelopes and intended to be representative of the variation in handwritten digits that might be encountered in the wild. *SVHN* is a widely used digits dataset and more challenging for classification tasks than *MN*, which consists of real-world images collected from house number plates. It contains over 70k $32 \times 32$ RGB images for training and over 20k for testing. The training set and test set for all these digit datasets are unbalanced, i.e., the number of each class is unequal, shown in the Tab. 6 and Tab. 7, respectively.

Table 6: The number of each class in the training set of the digits dataset.

|  | 0 | 1 | 2 | 3 | 4 | 5 | 6 | 7 | 8 | 9 |
|---|---|---|---|---|---|---|---|---|---|---|
| MNIST | 5923 | 6742 | 5958 | 6131 | 5842 | 5421 | 5918 | 6265 | 5851 | 5949 |
| USPS | 1194 | 1005 | 731 | 658 | 652 | 556 | 664 | 645 | 542 | 644 |
| SVHN | 4948 | 13861 | 10585 | 8497 | 7458 | 6882 | 5727 | 5595 | 5045 | 4659 |

Table 7: The number of each class in the test set of the digits dataset.

|  | 0 | 1 | 2 | 3 | 4 | 5 | 6 | 7 | 8 | 9 |
|---|---|---|---|---|---|---|---|---|---|---|
| MNIST | 980 | 1135 | 1032 | 1010 | 982 | 892 | 958 | 1028 | 974 | 1009 |
| USPS | 359 | 264 | 198 | 166 | 200 | 160 | 170 | 147 | 166 | 177 |
| SVHN | 1744 | 5099 | 4149 | 2882 | 2523 | 2384 | 1977 | 2019 | 1660 | 1595 |

We also apply the same data processing to all digits datasets, i.e., resizing all images to $32 \times 32$. We also apply the gray-scale transform to images in SVHN to be consistent with MNIST and USPS. As the visualization of each digit dataset shown in the Fig. 4, the SVHN dataset is more complex than the the MNIST and USPS datasets, making the classification task more challenging. Therefore, the model trained on SVHN will learn more complicated feature representation and can be transferred to the MNIST/ USPS domain with high accuracy.

**Natural Image**. Both CIFAR-10 and STL-10 have 10 classes. For CIFAR-10, it contains 60k $32 \times 32$ color images in 10 classes, with 6k labeled images per class. It is divided into 50k training images

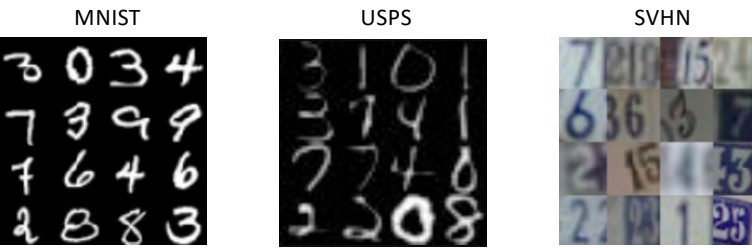

Figure 4: The visualization of the digits datasets.

and 10k test images. As for STL-10, it has 5k training images and 8k test images with the original size of 96×96. To make the DNN models applicable to both datasets, we down-scale the images to 32×32 resolution to match that of CIFAR-10.

## C DNN ARCHITECTURES

The VGG-11 architecture follows the setting from ConvNet Configuration A in Simonyan & Zisserman (2015) with a liitle modification. Specifically, each convolutional layer is followed by a batchnorm layer and ReLU operation. We also add ReLU and dropout after the first and the second linear layer.

The setting of ResNet-18 is the same as the one defined in He et al. (2016) except the kernel size of the first convolutional layer is set to 3×3 instead of 7×7. Similarly, we add batchnorm and ReLU after each convolutional layer.

Besides, for the victim model trained on the digits datasets, the input channel of the first convolutional layer is set to 1 for both DNNs, while it set to 3 for CIFAR dataset.

## D EXAMPLES OF AUXILIARY DOMAINS

We provide a few examples of auxiliary domains generated in Fig. 5 and Fig. 6. We also measure the MMD between the source domain and its auxiliary domains, and we observe the MMD increases with the value of $\mu$ when $\sigma$ is fixed. Specifically, in Fig. 5, the MMD is [0.017, 0.036, 0.041] when $\mu$ =[0, 0.25, 0.5] and $\sigma$ = 0.5.



$\mu = 0, \sigma = 0.5$      $\mu = 0.25, \sigma = 0.5$      $\mu = 0.5, \sigma = 0.5$

Figure 5: The examples of auxiliary domains for MNIST

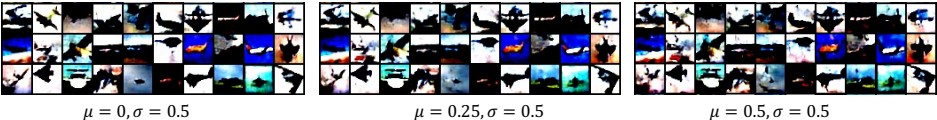

$\mu = 0, \sigma = 0.5$      $\mu = 0.25, \sigma = 0.5$      $\mu = 0.5, \sigma = 0.5$

Figure 6: The examples of auxiliary domains for CIFAR10

Besides, we use t-SNE to visualize the latent space of the source domain and the auxiliary domains, as shown in Fig. 7. Here we sample 100 data from 6 auxiliary domain for better visualization, where $\mu$=[0, 0.5, 1] and $\sigma$=[0.5, 1]. The Fig. 7 successfully demonstrates that the auxiliary domains can augment the source domain.

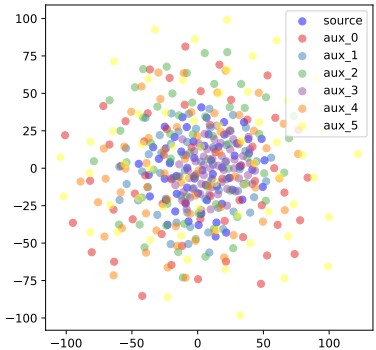

Figure 7: The visualization of of MNIST and its auxiliary domains using t-SNE

## E  HYPER-PARAMETERS

After attackers extract the pre-trained model, they will either directly use it in the same domain, or apply transfer learning with 1000 data in the target domain to improve the accuracy by fine-tuning the whole model. To mimic the real-world behavior of both attacker and protector, we construct a small searching space with common choices [0.01, 0.005, 0.001, 0.0005. 0.0001] for each side, and finally determine the initial learning rate for the protector is 0.0005, and for the attacker is 0.001. The learning rate will then decay at a rate of 0.5 every 10 epochs.

For the fine-tuning training strategy, we follow the same setting in Sec. 5.1, and the training process stops when the inference accuracy does not improve for the last 3 validation epochs.

## F  RESULTS ON CIFAR10 & STL10

Here we provide the protection performance of CIFAR10 & STL10 when attackers fine-tune the protected model for cross-domain transfer. The methods *NTL* and *CUTI* only experiment with VGG-13 on CIFAR10 & STL10, so we simply include their results as a comparison. As results shown in Tab. 8, through the modification of critical weights, DeTrans effectively reduces transfer accuracy from 82.95% to 10.87% for the STL10 to CIFAR10 transfer with VGG-11. In contrast, the SOTA methods only degrade accuracy to ∼14%. Additionally, the transferability of ResNet-18 under DeTrans protection is also reduced by up to 64.43%. The success of DeTrans can be attributed to the optimization of critical weights achieved through the bi-level optimization process.

Table 8: The results of defending against cross-domain transfer on natural image datasets. The left of ⇒ is the source domain, and the right side is the target domain (unseen during protection).

|  |  | CIFAR10 ⇒ STL10 | STL10 ⇒ CIFAR10 | Avg. Drop |
|---|---|---|---|---|
| | Baseline | 62.04 | 82.95 | / |
| | SL | 11.25 | 42.61 | 45.56 |
| VGG-11 | NNSplitter | 61.48 | 82.65 | 0.43 |
| | **DeTrans** | 10.35±0.56 | 10.82±0.79 | **61.91** |
| VGG-13 | NTL | 14.8 | 14.9 | / |
| | CUTI | 14.0 | 13.3 | / |
| | Baseline | 63.14 | 86.83 | / |
| | SL | 11.67 | 42.93 | 47.68 |
| ResNet-18 | NNSplitter | 62.37 | 86.14 | 0.73 |
| | **DeTrans** | 10.91±0.43 | 10.19±0.67 | **64.43** |

## G  ANALYSIS OF NAIVE OPTIMIZATION

In the application of naive optimization can lead to a back-and-forth process between the protector and the attacker. Initially, the protector modifies critical weights in the pre-trained model to degrade its accuracy in the target domain. Subsequently, the attacker extracts the model and performs fine-tuning

to improve its accuracy. This iterative exchange resembles a strategic online game, where each side responds to the actions of the other.

The resulting effect is depicted in Figure 8, where both sides experience drastic and non-converging changes in their losses, even when extending the training epoch from 50 to 100. In contrast, the bi-level optimization approach implicitly considers the attacker's actions, strategically thinking ahead by anticipating future steps, leading to improved performance.

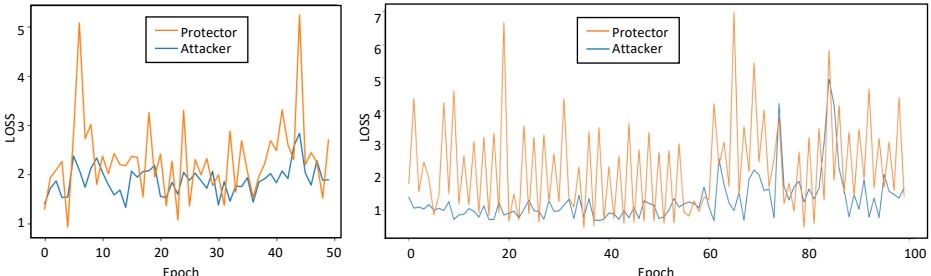

Figure 8: The loss of naive optimization.

