# OpenReview forum: "Two Birds with One Stone: Protecting DNN Models Against Unauthorized Inference and Domain Transfer"
_ICLR.cc/2024/Conference — Submitted to ICLR 2024_

### Official Review · Reviewer_a4Uq · 2023-10-29

**Soundness:** 3 good
**Presentation:** 3 good
**Contribution:** 3 good
**Rating:** 5
**Confidence:** 2

**Summary:**

In this paper, the authors propose the first work that mitigates the risk of model leakage by preventing attackers from both unauthorized inference and cross-domain transfer, thus achieving comprehensive protection for DNN models. Experiments demonstrate their design outperforms the state-of-the-art model protection works and exhibits robustness against different fine-tuning methods employed by attackers.

**Strengths:**

- Propose the first work that mitigates the risk of model leakage
- Experiments demonstrate their design outperforms the state-of-the-art model protection works

**Weaknesses:**

- The threat model is not practical

In this paper, the authors assume an unpractical threat model for the attacker. Although this can be the worst case for the defender, such unpractical threat model for attack can make that such an attack can never happen in the real world. Thus, developing a valid defense for such unrealistic attack is not very meaningful. Thus, it would be great if the authors can provide more detailed discussion or justification on the threat model side.

- Lack of justification on the representativeness of their evaluation setup

The evaluation setup for this paper such as dataset and models selection is very ad-hoc. There is no justification on the reason for such selections. For instance, are the selection representative or the state-of-the-art? Without such justification, it is unclear whether their designs and findings can transfer and work well in the most representative setup.

**Questions:**

Provide the justification on the threat model and evaluation setup.

---

> ### Author Response · Authors · 2023-11-15
>
> Thanks for your comments.
>
> **[For Q1]**
>
> We appreciate your feedback regarding the practicality of our threat model. To address your concern effectively, **we would like to understand which specific aspect you find impractical.** It's crucial for us to receive more technical details/comments to provide a targeted clarification and justification for our chosen threat model. **It's worth noting that Reviewer rjhp considered our threat model realistic and listed it as a strength.** We are eager to ensure a clear understanding and address any potential concerns regarding the practicality of our threat model.
>
> **[For Q2]**
>
> Indeed, we have provided detailed justifications for our choices in Sections 4.1 and 4.2, **aligning with state-of-the-art works** to ensure fair and meaningful comparisons.
>
> **Given your positive evaluation of the robustness, presentation, and contribution of our work, and recognizing the impressive results we achieved, we respectfully ask for a reconsideration of the rating in light of the information we provided (including responses to other reviewers if helpful).**  We firmly believe that our research makes a substantial contribution to enhancing on-device ML protection by effectively tackling dual vulnerabilities, with the results showcasing the effectiveness of our proposed approach.

---

### Official Review · Reviewer_RU2E · 2023-11-01

**Soundness:** 3 good
**Presentation:** 3 good
**Contribution:** 3 good
**Rating:** 5
**Confidence:** 4

**Summary:**

**Paper Summary**

This paper presents a method to protect the intellectual property (IP) of a model when an attacker gains access to both the model architecture and parameters. The approach employs a bi-level optimization method to reduce the fine-tuning accuracy of a model on various nearby target domains. The results show that this approach can significantly reduce the model transferability from the source domain to any unauthorized target domain.

**Strengths:**

**Strengths:**
- The paper is well-organized and easy to follow.
- The algorithm is robust.

**Weaknesses:**

**Weaknesses:**
- I have concerns regarding the threat model and evaluation method which are given in questions.
- The practical real-world applications of this approach are not clear.

**Questions:**

**Comments:**
- I'm uncertain if this is a mistake, but in your threat model, you mentioned, "We assume the attacker is able to extract the DNN model, including its architecture and well-trained weights." I find this confusing. What exactly is the Trusted Execution Environment (TEE) used to protect in this context? How is it feasible to protect the model against **unauthorized inference** when the attacker has both the model weights and architecture?

- Comparing transferability is challenging. In the evaluation, you present the final test accuracy of a model transferred from the source to the target domain, which is a good starting point. However, it is hard to conclude that the approach reduces model transferability. Many other factors could influence your evaluation, such as model initialization and target domain selection. Is it possible to include learning curves in your analysis?

- Can you show that the bi-level optimization added during the training stage does not compromise model quality in the source domain? What you are trying to show is the existence of distinct local minima for the source domain compared to nearby target domains. This is very hard to prove.

I find the problem and the solution in this paper intriguing. However, I still have some concerns regarding the evaluation and threat model.

---

> ### Author Response · Authors · 2023-11-15
>
> Thanks for your valuable comments.
>
> **[For Q1]**
>
> Our work aims to ensure the secure deployment of pre-trained models by generating a protected version. **The protected model is deployed on devices, and crucial data (the original values of the perturbed weights) is safeguarded by TEE**. During inference, users can access this data but cannot extract it, thanks to the properties of TEE. On the other hand, attackers cannot even gain access to the secure memory of TEE, although they can extract the unprotected model deployed in unsecured memory, which exhibits low performance for both domains. Therefore, our assumption is that the protected model can be white-box for attackers, which is a stronger threat model with a stronger attacker. We will provide further clarification on the threat model and delve into the details of TEE properties in the final version to eliminate any confusion.
>
> **[For Q2]**
>
> Our method takes pre-trained models as input and generates their protected versions. To avoid introducing any initialization bias, we utilized open-sourced pre-trained models available online. For the selection of the target domain, we adhered to the experiment settings outlined in SOTA works [1] and [2]. Notably, these references did not include the learning curve in their work, thereby we overlooked its significance. However, we appreciate your suggestion and are willing to incorporate it into the final version.
>
> **[For Q3]**
>
> The primary advantage of our method lies in leveraging TEE to ensure model quality on the source domain for users. Specifically, we perturbed one filter per layer to generate the protected model through bi-level optimization. The original values of these perturbed weights are stored inside the TEE to preserve the source performance for users. Consequently, **bi-level optimization does not compromise the model quality in the source domain.**
>
> We acknowledge that the way we leverage TEE may cause confusion to you. We hope the information provided above addresses your concerns.
> We appreciate your acknowledgment that our paper is intriguing and hope we have addressed all your concerns.
>
> ---
> Reference:
>
> [1] Non-transferable learning: A new approach for model ownership verification and applicability authorization. In International Conference on Learning Representations (ICLR), 2022.
>
> [2] Model barrier: A compact un-transferable isolation domain for model intellectual property protection. In Proceedings of the IEEE/CVF Conference on Computer Vision and Pattern Recognition, pages 20475–20484, 2023.

---

### Official Review · Reviewer_rJhp · 2023-11-01

**Soundness:** 3 good
**Presentation:** 3 good
**Contribution:** 3 good
**Rating:** 3
**Confidence:** 4

**Summary:**

This work presents DeTrans which protects on-device DNN models against unauthorized inference and cross-domain transfer while preserving model performance for users using TEE.  By selectively modifying a small subset of weights in the pre-trained model, DeTrans achieves near-random guess performance on the source domain and transferability reduction for potential target domains.

**Strengths:**

Threat model covering both unauthorized inference and cross-domain transfer is realistic.

**Weaknesses:**

1. Evaluation is limited to small dataset. Can authors provide results on CIFAR-100, Tiny-ImageNet and ImageNet?

2. Lack of comparisons against wider range of prior arts.

**Questions:**

Please see the weakness

---

> ### Author Response · Authors · 2023-11-15
>
> Thank you for your feedback, and **we appreciate your acknowledgment of the soundness, presentation, and contribution of our work.** We understand your interest in broader evaluations and comparisons.
>
> As mentioned in Section 4.2, our work is pioneering in simultaneously addressing both unauthorized inference and cross-domain transfer (i.e., dual protection). Given the novelty of our approach, there are no previous works that are directly comparable or as the baseline. However, we tried our best to find comparison methods, including state-of-the-art approaches like [1-3]. Moreover, for fair evaluation, we intentionally aligned with their datasets and model selections to ensure meaningful comparisons and to underscore the notable performance achieved by our method.
>
> The choice of datasets in our evaluation **aligns with the common settings in state-of-the-art works focusing on cross-domain transfer**, where the source and target domains share the same label space. To the best of our knowledge, **large datasets like CIFAR-100, Tiny-ImageNet, and ImageNet are not commonly used in these works and ours, likely due to the challenge of constructing appropriate target domains.** We are open to incorporating your suggestions if you can guide us to cross-domain transfer works that use these larger datasets, allowing us to follow similar settings.
>
> Regarding a wider range of prior arts, **we have made efforts to include relevant and state-of-the-art methods for comparison.** However, **if you have specific suggestions for additional prior arts, we would appreciate your insights.**
>
> **Considering your positive assessment of our work's soundness, presentation, and contribution, and acknowledging that the evaluation is your primary concern, we kindly request you to reconsider the rating in light of the information provided above. We believe our work significantly advances on-device ML protection by addressing dual vulnerabilities, and the results demonstrate the efficacy of our proposed method.**
>
> ---
> Reference:
>
> [1] NNSplitter: An active defense solution for DNN model via automated weight obfuscation.
> In Proceedings of the 40th International Conference on Machine Learning, 2023.
>
> [2] Non-transferable learning: A new approach for model ownership verification and applicability authorization. In International Conference on Learning Representations (ICLR), 2022.
>
> [3] Model barrier: A compact un-transferable isolation domain for model intellectual property protection. In Proceedings of the IEEE/CVF Conference on Computer Vision and Pattern Recognition, pages 20475–20484, 2023.

---

> > ### Comment · Reviewer_rJhp · 2023-11-22
> >
> > NNSplitter presents results on CIFAR-100.

---

> > > ### Author Response · Authors · 2023-11-23
> > >
> > > It is crucial to highlight that **NNSplitter did not encompass the aspect of unauthorized cross-domain transfer**. In contrast, our research accounts for dual vulnerabilities, necessitating experimental settings that can be applied to both direct usage and cross-domain transfer scenarios. It is noteworthy that the latter is not carried out on datasets such as CIFAR100 or ImageNet, to the best of our knowledge.
> > >
> > > We wonder if you have any further constructive suggestions, e.g., **regarding our methodology** (which is the main contribution we want to make to the field). **If the only concern is the scale of our experiments, we believe our work is understated, particularly given our explanation that we followed the settings of state-of-the-art works.**

---

> > > > ### Comment · Reviewer_rJhp · 2023-11-23
> > > >
> > > > Thank you for your prompt response. I appreciate your effort to ensure a fair comparison with prior work, which is commendable. However, when claiming to be the first in a particular area of research, it is imperative to establish a robust and appropriate baseline for future studies. While comparison with previous methods is essential, it is equally important to present thorough and non-comparative evaluations of your own method to demonstrate its efficacy.
> > > >
> > > > You are looking for 'source and target domains share the same label space', which is hard. However, does this indicate that your assumption is not so realistic (or the cross-domain transfer tasks in general are not so realistic?)
> > > >
> > > > For larger dataset, have you considered dividing a single dataset into two segments to test your method, even though this may simplify the attack due to the close distribution? Can you select the overlapped (or semantic similar) labels in different datasets and conduct your method?
> > > >
> > > > 'NNSplitter did not encompass the aspect of unauthorized cross-domain transfer', can you just compare with it in terms of unauthorized inference on CIFAR-100?
> > > >
> > > > Furthermore, adapting your method to attention models could significantly enhance its applicability.
> > > >
> > > > In summary, based on the current experimental results presented in the paper, the applicability of the proposed method appears quite limited, particularly regarding model size and dataset variety. Consequently, I think the overall impact of the work may not meet the high standards expected for acceptance at ICLR. I encourage a broader and more in-depth exploration of the method's potential applications and effectiveness across various scenarios to strengthen the paper's contribution to the field.

---

### Official Review · Reviewer_Uryd · 2023-11-06

**Soundness:** 2 fair
**Presentation:** 1 poor
**Contribution:** 2 fair
**Rating:** 3
**Confidence:** 5

**Summary:**

The paper  considers the problem of preventing misuse of DNNs from an IP perspective in two ways:
(1)  prevent unauthorized users or use on unauthorized hardware from using the model; In past work, this problem has been addressed by use of TEEs. The model is protected in some way, with various approaches, in a hardware-based TEE. Any use on unlicensed hardware either results in degraded performance or entirely prevented.

(2)  Even if (1) is ensured, restrict the use of the model to only the source domain (a specified domain) and not allow use on other domains. This is referred to as cross-domain transfer. As the paper on page 1 points out, (Wang et al. 2022/2023) identified this problem.

Unfortunately, what is not clear is why prior work doesn't already solve the problem.  See Questions below in the review. It appears that (Wang et al. 2022/2023) not only identified the problem, but also proposed a solution to the problem.  Furthermore, Wang et al. 2022 cites to solutions for (1) and was meant to address the gap (2), and thus anticipated the straightforward combination.  In other words, One uses Wang et al. to develop a model that performs well on the source domain but poorly on non-source domains and then uses a solution to (1) to restrict the availability of the resulting model on only authorized hardware.  Thus, the motivation for the work is not clear and the premise appears weak or lacking a clear well-stated motivation.

The paper  points to Wang et al. 2022 again in Section 2.4,   acknowledging that they solve (2), but then says these methods cannot be extended to "pre-trained models".  But, if this limitation of Wang et al. is really important, it seems that should have been introduced in the abstract and Intro and claimed as the contribution. But the paper does not do that.

Overall, it is not clear precisely what problem is being solved in this paper that is not already solved. The abstract and Intro need to explain the contributions and the problem much better. For instance, if the paper is really addressing that the model needs to be extensible to "pre-trained models", and that is the distinguishing contribution from prior work, the overall pitch needs to change considerably.

**Strengths:**

The paper correctly points out that DNNs can be valuable IP and limiting their use to authorized devices and only on desired domains (source domains) are important. It provides a framework that claims to address both issues.

**Weaknesses:**

The paper lacks proper foundation. It seems the problems that the paper claims to solve for the first time are already solved by prior work or easily addressed by combining existing techniques. See Questions below.

I recommend addressing that first (see Questions below). Once that is addressed, I may have further questions on the performance results and the proposed method. But, right now, the contributions are not properly situated with respect to prior work and addressing that is crucial.

 The rebuttal suggests that authorized users will not do any attacks and are fully benign. Is that a realistic assumption? Can't an unauthorized user simply become an authorized user, if the device is cheap to buy (it seems the authors assume the devices are relatively modest devices). It would be good to know one scenario where the threat model is realistic. If the distribution of the device is extremely restricted, is the solution needed?


It appears that the scheme is a variant of Zhou et al.'s  2023 scheme, but weakens some of the security assumptions in that paper (e.g., preventing recognition of model parameters that are obfuscated by the attacker).  That should be clearly acknowledged, if some security assumptions are  being given up or have to be given up. Couldn't the same security assumptions be retained and the scheme built along the line of Zhou et al.s 2023, but with a different training objective?  This weakens the soundness of the scheme from a security perspective with respect to (Zhou et al. 2023). The paper does not evaluate that or consider that or even acknowledge that.

Adaptive attacks are not considered or acknowledged.

**Questions:**

I would like to see the weaknesses addressed.

 I would like to see more clarity on the premise of the paper (motivation).

The paper should clarify why  (Sun et al. 2023) prevents unauthorized use cannot be combined with Wang et al. (2022, 2023) and achieve the desired goals (or, for that matter, Wang et al. 2022/2023 objective function combined with Zhou et al. 2023's approach of restricting the changes to a small set of weights).

(I have updated the review after an extensive back-and-forth with the authors.)

---

> ### Author Response · Authors · 2023-11-15
>
> Thank you for your feedback, and we appreciate the opportunity to address your concerns and clarify certain aspects of our work.
>
> **[Terminology Clarification]**
>
> First, we would like to clarify **the misunderstanding between the terms "well-trained models" and "pre-trained models" that were used interchangeably** in this paper, to convey the same meaning. We **did mention "well-trained" models several times in the introduction section (and in the threat model)**, to emphasize that our research objective in this work is to securely deploy well-trained/pre-trained models.
>
> Following your suggestion, we will consistently use "pre-trained models" to avoid ambiguity. Additionally, we will explicitly include the keyword "pre-trained models" in the abstract and contribution sections for better clarity.
>
> **[Motivation]**
>
> We have to emphasize that **our work uniquely identifies a security problem related to on-device deep neural networks (DNNs)** — specifically, the dual vulnerabilities associated with preventing unauthorized use and restricting cross-domain transfer. The solution we proposed **cannot be simply replaced by the combination of current works**, for the following reasons.
>
> First, in fact, **Wang et al. 2022 [1] did not anticipate combining the two approaches**, since their threat model is based on AIaaS, where the model providers have no motivation to degrade the performance on the source domain. However, **our threat model is totally different, and based on on-device ML**, i.e., we are targeting different problems in this work.
>
> Besides, **the suggestion to combine Chakrabory et al. 2020 [2] and Wang et al. 2022 [1] is infeasible**, primarily because they employ different training strategies and start from scratch, i.e., model owners cannot first use one method to train the model then use another one to train again and expect the resulting model can address two vulnerabilities at the same time.
>
> **The motivation for our work lies in providing a comprehensive defense solution that extends to pre-trained models, granting model owners the flexibility to preserve valuable model properties**, such as transferability. This flexibility is a key advantage. To elaborate, model owners can retain the pre-trained model and efficiently transfer it to other domains, if needed. Simultaneously, they can employ our method to securely deploy the model on devices. **In contrast, directly adopting the approach proposed in Wang et al. 2022 [1] would confine the model to the domains it was initially trained on, limiting both its flexibility and transferability, even for the model owners themselves.**
>
> We will incorporate the above discussion into the final version to provide a more detailed elaboration of our motivation.
>
> ---
> Reference:
>
> [1]  Non-transferable learning: A new approach for model ownership verification and applicability authorization. In International Conference on Learning Representations (ICLR), 2022
>
> [2] Hardware-assisted intellectual property protection of deep learning models. In 2020 57th ACM/IEEE Design Automation Conference (DAC), pp. 1–6. IEEE, 2020.

---

> ### Author Response · Authors · 2023-11-15
>
> **[Threat Model]**
>
> We utilized Sun et al. (2021) [3] as an example to illustrate the practical vulnerability of model extraction, highlighting the white-box nature of the threat. Although various papers, including Sun et al. (2023) [4], have proposed solutions leveraging TEE to mitigate model extraction, the critical challenge of universally employing TEE for DNN model protection still exists. Notably, Sun et al. (2023) [4] has its own limitations, as already discussed in [5]. Importantly, compared to Sun et al. (2023) [4], our work addresses a more challenging security problem, encompassing both model extraction and unauthorized domain transfer in on-device ML scenarios, and we employ TEE in a novel manner to tackle this dual threat.
>
> **[Clarification for Q3]**
>
> The assumption that attackers are able to extract the model weights completely **doesn't violate the requirement** that unauthorized inferences on the source domain must be prevented. The reason is that **we deployed the protected model on the device instead of the pre-trained/victim model, as clearly stated in the introduction**:
>
> > Therefore, when deploying the protected model on the device, attackers cannot achieve high performance on both domains, thus achieving double protection. Meanwhile, the original value of the perturbed weights is small enough to be stored in the stringent secure memory of TEE, which can be used to restore high performance on the source domain for users.
>
>
> Besides, **your premise is problematic since the un-obfuscated model weights cannot become available to attackers**. We explicitly mentioned that we use TEE to secure these data in the Introduction, Sections 2.1, and 2.3. This reliance on TEE security is fundamental to our approach, which we assume is the basic background for a reviewer like you (given that your review confidence is quite high).
>
> **[Clarification for Q4]**
>
> The challenge of an unknown target domain is a common concern addressed by various works, and **we did not claim it to be a unique challenge**.
>
> Specifically, **we do not agree with the reviewer that addressing a research challenge loses significance just because there have been prior attempts, even if they are not applicable or offering high-performance**. Even for Wang et al. 2022 [1] as you mentioned, it is not the initial work to tackle the challenge of an unknown target domain, as demonstrated by prior work in single-domain generalization, such as [7]. Still, it makes great and unique contributions to solve a different problem. Also, Wang et al. 2022 [1] has limitations as discussed in [6], which tackled the exact same challenge using a different method. We did not thoroughly discuss such limitations since **how to tackle an unknown target domain is not our focus or the main contribution we want to make**.
>
> **Our focus is on providing a comprehensive defense solution, extending to pre-trained/well-trained models for on-device ML. The gap between our work and prior approaches is discussed to highlight the specific contributions and advancements of this work.**
>
> ---
> Reference:
>
> [3] Mind your weight (s): A large-scale study on insufficient machine learning model protection in mobile apps. In 30th USENIX Security Symposium (USENIX Security 21), pp. 1955–1972, 2021.
>
> [4] Shadownet: A secure and efficient on-device model inference system for convolutional neural networks. In 2023 IEEE Symposium on Security and Privacy (SP).
>
> [5] Teeslice: slicing dnn models for secure and efficient deployment. In Proceedings of the 2nd ACM International Workshop on AI and Software Testing/Analysis, pages 1–8, 2022.
>
> [6] Model barrier: A compact un-transferable isolation domain for model intellectual property
> protection. In Proceedings of the IEEE/CVF Conference on Computer Vision and Pattern Recognition, pages 20475–20484, 2023.
>
> [7] Learning to learn single domain generalization. In Proceedings of the IEEE/CVF Conference on Computer Vision and Pattern Recognition, pages 12556–12565, 2020.

---

> ### Comment · Reviewer_Uryd · 2023-11-21
>
> I would like to see the revised pitch (Abstract and Intro at least) that restates and clarifies the contributions.
>
> I am not convinced why Wang et al 2022 cannot be combined with a TEE-based approach. Doesn't it provide a valid defense? I get your argument that Wang et al. 2022 may require retraining if the original owners want to transfer the model -- but that doesn't make the combo invalid -- only less efficient. Can you elaborate in more detail on why they can't be combined? There are multiple TEE-based approaches, including the more recent NNSplitter work. None of them can be combined with each other?
>
> It seems to me that if they can be combined, even if that requires additional training, one is largely left with an efficiency argument. (Improved efficiency can still be a valid contribution -- but the paper's pitch has to be consistent with that, acknowledging that there is a different solution strategy.) If you are claiming that a combination is simply not possible for the two threads of research in the past, then I think that needs a much stronger technical argument than currently made in the paper to convince a reader. Either way, it seems to me that the paper's thrust needs to be revised significantly, with a much clearer articulation of the claimed contribution over past work.

---

> > ### Author Response · Authors · 2023-11-21
> >
> > We have revised the pitch. Please check the updated version. Thanks for your time.

---

> > > ### Comment · Reviewer_Uryd · 2023-11-23
> > >
> > > One more concern. If the threat model for transfer attacks to a nearby domain is that the attacker is an unauthorized user (which I think should be specified clearly), then there are two possible issues:
> > >
> > > 1. Can the attacker simply become an authorized user by purchasing the device (not sure how expensive that is) and then do model stealing? This also breaks NNSplitter, I think.   It seems there  is an unstated assumption in all the work that somehow becoming an authorized user is very expensive or queries are very expensive for an authorized user so that model with true weights cannot be stolen. In what settings is that realistic?
> > >
> > >
> > > 2. Did you consider the attacks that were mentioned in NNSplitter? For instance, they tightly bounded the modified weights so that the attacker could not easily figure out which weights were perturbed. If they could figure that out, then they could potentially correct them even with small datasets.  In your case, it is not clear if all the defenses that were deployed in NNSplitter are also deployed. Else, you could be vulnerable to some of the attacks they mentioned.
> > >
> > > Thanks.

---

> ### Comment · Reviewer_Uryd · 2023-11-23
>
> Thanks for updating it. I think the paper is improved, but I don't think the paper is quite there yet.
>
>  I moved my score up from my original score accordingly to a reject from strong reject, but I  think the paper still has issues in clarity of contributions over prior art. You may also want to fix the incomplete sentence just before Section 3 -- I think you were trying to make some argument, but it was left midway.
>
> A few more thoughts as I read the updated paper for you to consider:
>
> 1. My understanding is that the cross-domain transfer you are worried about is by authorized users. For unauthorized users, it seems the model performance is anyway degraded by techniques such as (Zhou 2023).  If so, it would have been clearer to state that in the Intro clearly.
>
> 2. My understanding is that you try to prevent the transfer attacks by authorized users to domains that are "close" to the source domain. Closeness is defined by a distance threshold. Could an adversary first transfer the model to a domain that is just outside that threshold? And then perhaps at a much lower cost, retransfer it to a close function? I am thinking of an adaptive adversary here.
>
> 3. On-device versus MLaaS argument: How critical is that to your paper to distinguish from prior work? If it is critical, I think you need to also argue why prior work could not simply be run on-device rather than in MLaaS mode. Is there something fundamental that prevents Wang et al. (2022, 2023) from being used in on-device settings? Regardless, it appears NNLsplitter work (Zhou 2023)  assumes on-device as well.
>
> 4. The approach in section 3.2 seems to be inspired by   Zhou 2023 (NNSplitter). They also modified a small set of weights, but to degrade unauthorized use. I wonder if the paper would have read better if throughout the paper, including the Intro, Zhou 2023 was used as the closest work to your work (acknowledging that). For instance, Zhou et al.  addressed  the on-device model extraction issue by unauthorized users, but not addressing domain transfer by authorized users. You could also point out why earlier work prior to Zhou et al. were more limited in scope -- e..g, required a  much larger-memory TEE or could not be used on-device (and why).
>
> I think scoping the problem correctly with proper context would have gone a long way in improving the paper.
>
> 5. Now coming to the question of preventing model transfer by authorized users to a nearby domain in the on-device scenario, it is not clear if  you considered the following attack strategy:   An authorized user simply steals the source model by using a data-free method such as MAZE that does not require source data. It seems to me that most of the queries with systems such as MAZE would be far away from the source domain and thus not disabled by the proposed defense, since it only addresses queries from a nearby distribution.  In the on-device scenario, it seems to me that data-free model stealing attacks should  be easier since the attacker has local, fast access to the on-device model.
>
> Note that in Zhou 2023, they assumed that the attacker (an unauthorized user in their case)  can completely steal the model, including exact weights. Of course, in their scheme, the attacker would only get a model with many weights incorrect. They further assumed that the attacker does not have source data (or very limited source data) -- and thus cannot retrain a model. Thus, there were fewer concerns under their threat model, since authorized users were considered trusted. In your threat model, authorized users are not so trusted -- they may attempt to transfer the model to nearby domains. So you do need to consider stronger attacks.
>
> Thus, in your case, under the same assumptions,  it appears the authorized user can completely extract the source model.  A possible next step to bypass your defense would be to  then  transfer the model (by some retraining).  This is model extraction followed by transfer, rather than transfer followed by model extraction that is assumed in your paper.
>
> Overall, I think the paper writing has to improve with more thought given to the pitch, why it really differs from prior work, and why adaptive attacks on your defense may not work (or you may find that they will work, in which case you can write an attack paper showing that no defense that allows model stealing on the source domain can prevent model transfer.) A more comprehensive analysis of potential attacks on your defenses may be needed.
>
> In summary, consider improving the paper significantly by sharpening the pitch.  It could be that it is a better fit for a security-oriented conference.   I recommend improving the work, thinking a bit more deeply on the defense as well as potential adaptive attacks, and resubmitting to an appropriate venue. Best wishes.

---

> ### Author Response · Authors · 2023-11-23
>
> Thanks for pointing out the unfinished sentence and we fixed it in the newly uploaded version (which also clarifies the threat model and introduces more details about TEE for better understanding) .
>
> **[For 1]** Your understanding is incorrect. In our work, the authorized users are attackers who would extract the model from unsecured memory and use it for direct use or cross-domain transfer.
>
>
>
> **[For 2]** If you are considering a more adaptive adversary, at least you agree that our threat model is reasonable and that our work has addressed the vulnerability we identified.  Even if the adaptive attack you briefly mentioned can be a real attack (although we do not how this attack exactly happened), you should recognize that our method has already increased attackers' efforts in obtaining a useful model. This is sufficient to demonstrate the significance of our defense work.
>
> **[For 3]** Given your high confidence, it should be clear that on-device ML and MLaSS face different vulnerabilities ---  We have explained there is no motivation for MLaSS works to degrade performance on the source domain, but we are motivated to do it as a defense method (while leveraging TEE to ensure performance for authorized users). Also, While we acknowledge that NNSplitter is designed for on-device ML, as discussed in the Introduction and related work (Sec. 2.4), it only considers one vulnerability.
>
> **[For 4]** As mentioned above, we clearly mentioned NNSplitter at the very beginning (the first paragraph in Intro), also discussed in Sec 2.4.
>
> **[For 5]** In our threat model, authorized users are benign users (e.g., those who bought our ML application) and will not conduct model-stealing attacks, whereas attackers are unauthorized users who did not pay for the application but want to extract it and use it for free (either direct use or do cross-domain transfer). Also, our threat model is quite similar to NNSplitter, with the difference being that we consider attackers could also conduct cross-domain transfer.
>
> Additionally, even for authorized users, they cannot extract the data inside TEE, which remains a black box to them (according to the property of TEE). We added this background in the revision.
>
> We observe that our main discussions revolve around different concepts,  unproposed combinations of prior works, or briefly mentioned attacks that may be unpractical. However, we regret that there is no opinion on our methodology/experiment/performance. We hope our response can give you a better understanding of our work so that you may point out any technical problems for us to improve the work. With only conceptual concerns (which do not bother any other reviewers), we can only assume that it is just our storytelling that does not fit your taste, but there's no indication of anything wrong with other parts.
> We respect your decision if you are reluctant to increase the score, but we would appreciate more constructive suggestions to ensure that we are convinced your rating is fair.

---

> ### Author Response · Authors · 2023-11-23
>
> For 1, the threat model you mentioned is different from ours. We are pleased to note that our research has inspired others, such as yourself, to delve into more adaptive attacks or consider different threat models.
>
> For 2, yes, we did consider.
>
>
> We find it difficult to align with your ratings and justification, especially considering the poor (score 1) ratings for Soundness, Presentation, and Contribution, while all other reviewers have given good (score 3) ratings. Furthermore, considering your self-assessed highest confidence, we are perplexed as to why certain basic aspects of on-device ML and TEE seem unclear to you (while our work is leveraging TEE to mitigate vulnerabilities in on-device ML).
>
> Lastly, we want to emphasize that it is challenging for a research paper to address all problems in a single endeavor, particularly in the realm of security papers, which often spawn follow-up papers proposing adaptive attacks or enhanced defenses. The initial exploration of a novel attack or defense in a paper sets the stage for subsequent researchers to propose more advanced solutions. Such contributions should be acknowledged, or at the very least, not downplayed.
>
> Your raised open questions (e.g., potential adaptive attacks or different threat models), some of which might be worth exploring for other researchers, should not diminish our contribution. On the contrary, they demonstrate that our work could lay the foundation for future endeavors.

---

> > ### Comment · Reviewer_Uryd · 2023-11-23
> >
> > Thanks for all your clarifications.
> >
> > I have updated my review, based on your inputs to my questions.

---

### Author Response · Authors · 2023-11-19
**To All Reviewers**

We would appreciate confirmation that we have sufficiently addressed all your concerns. Additionally, if you choose to maintain your current ratings after we provide further information or clarification, we kindly request justification for your decision. Thank you for your time.

---

> ### Author Response · Authors · 2023-11-21
>
> As the rebuttal period comes to a close, we genuinely look forward to receiving your response. Your time and consideration are highly valued. Thank you.

---

### Meta-Review · Area_Chair_4jtD · 2023-12-07

**Metareview:**

This paper aims to prevent on-device models from being reverse-engineered and then transferred to other domains.
The paper proposes a solution that combines trusted execution environments and optimization techniques to hinder transferability.

The ideas of protecting on-device models is certainly interesting, but the threat model and approach considered in this work have been quite confusing for all reviewers, and it is not quite clear what this paper achieves on top of prior approaches.

**Justification For Why Not Higher Score:**

The threat model considered here is somewhat convoluted. All reviewers had trouble understanding what the paper aims to achieve exactly, and how it differs from prior work.
I recommend the authors improve the presentation of the work to clarify the threat model and distinction from prior work.

**Justification For Why Not Lower Score:**

N.A

---

### Decision · Program_Chairs · 2024-01-16

Reject